

# Molecular phylogeny of mulberries reconstructed from ITS and two cpDNA sequences

Yahui Xuan, Yue Wu, Peng Li, Ruiling Liu, Yiwei Luo, Jianglian Yuan, Zhonghuai Xiang and Ningjia He

State Key Laboratory of Silkworm Genome Biology, Southwest University, Chongqing, China

## ABSTRACT

**Background:** Species in the genus *Morus* (Moraceae) are deciduous woody plants of great economic importance. The classification and phylogenetic relationships of *Morus*, especially the abundant mulberry resources in China, is still undetermined. Internal transcribed spacer (ITS) regions are among the most widely used molecular markers in phylogenetic analyses of angiosperms. However, according to the previous phylogenetic analyses of ITS sequences, most of the mulberry accessions collected in China were grouped into the largest clade lacking for phylogenetic resolution. Compared with functional ITS sequences, ITS pseudogenes show higher sequence diversity, so they can provide useful phylogenetic information.

**Methods:** We sequenced the ITS regions and the chloroplast DNA regions *TrnL-TrnF* and *TrnT-TrnL* from 33 mulberry accessions, and performed phylogenetic analyses to explore the evolution of mulberry.

**Results:** We found ITS pseudogenes in 11 mulberry accessions. In the phylogenetic tree constructed from ITS sequences, clade B was separated into short-type sequence clades (clades 1 and 2), and a long-type sequence clade (clade 3). Pseudogene sequences were separately clustered into two pseudogroups, designated as pseudogroup 1 and pseudogroup 2. The phylogenetic tree generated from cpDNA sequences also separated clade B into two clades.

**Conclusions:** Two species were separated in clade B. The existence of three connection patterns and incongruent distribution patterns between the phylogenetic trees generated from cpDNA and ITS sequences suggested that the ITS pseudogene sequences connect with genetic information from the female progenitor. Hybridization has played important roles in the evolution of mulberry, resulting in low resolution of the phylogenetic analysis based on ITS sequences. An evolutionary pattern illustrating the evolution history of mulberry is proposed. These findings have significance for the conservation of local mulberry resources. Polyploidy, hybridization, and concerted evolution have all played the roles in the evolution of ITS sequences in mulberry. This study will expand our understanding of mulberry evolution.

Corresponding author
Ningjia He, hejia@swu.edu.cn

## INTRODUCTION

Mulberries (*Morus* spp.), in the family Moraceae (order Rosales), are deciduous woody plants of great economic importance. The leaves of mulberry are the main food for silkworms (*Sánchez, 2002*). The fruit of *Morus* species also has nutritional and medicinal value (*Chen et al., 2016b*; *Priya, 2012*). Mulberry is believed to have originated in the Himalayan foothills and spread across the Eurasian, American, and African continents (*Nepal & Ferguson, 2012*; *Vijayan, Srivastava & Awasthi, 2004*). In 1753, Linnaeus assigned seven species in the genus *Morus* based on the color of aggregate fruits, leaf shape, and the presence of a cuticle (*Linnaeus, 1753*). Since then, many taxonomists have revised the classification of *Morus* (*Bureau, 1873*; *Koidzumi, 1917*; *Zhou & Gilbert, 2003*), and 10–16 species are currently recognized (*Nepal & Ferguson, 2012*; *Sánchez, 2002*). Spontaneous and artificial hybridization can occur between different species of mulberry, leading to successive interspecific characteristics (*Botton et al., 2005*). The continuous variations in the phenotypic characteristics have made it difficult to classify mulberry resources (*Vijayan, Srivastava & Awasthi, 2004*). Consequently, the classification of mulberry remains highly controversial.

Compared with phenotypic characters, DNA-based molecular markers represent a faster and more reliable system for germplasm characterization and phylogenetic analyses. Another advantage of DNA-based molecular markers is that they are not influenced by the environment (*Banerjee, Chattopadhyay & Saha, 2016*). Sequence-related amplified polymorphic, inter-simple sequence repeat, simple sequence repeat, and random amplified polymorphic DNA markers have been used for phylogenetic analyses of mulberry (*Banerjee, Chattopadhyay & Saha, 2016*; *Sharma, Sharma & Machii, 2000*; *Zhao et al., 2007*). In these studies, DNA markers were successfully used to identify mulberry accessions and analyze their genetic diversity. The results were consistent with classifications based on morphological characters (*Zhao et al., 2005*, *2007*). Internal transcribed spacer (ITS) regions (ITS1, 5.8S, and ITS2) are among the most widely used molecular markers in phylogenetic analyses of angiosperms (*Alvarez & Wendel, 2003*; *Baldwin et al., 1996*; *Li et al., 2011*) and are proposed to be core barcodes for seed plants (*Li et al., 2011*). These markers have also been used in phylogenetic analyses of mulberry. Based on ITS sequences and *trnL-trnF* sequences data of 13 mulberry accessions and *Broussonetia papyrifera*, Zhao divided *Morus* into five major clades, and identified *Morus* as a monophyletic group (*Zhao et al., 2005*). Nepal and Ferguson recognized 13 species of *Morus*, but phylogenetic analyses of ITS sequences and *trnL-trnF* sequences determined that this genus is not monophyletic, compared with 12 species in another genus (*Trophis*, *Bagassa*, *Milicia*, *Sorocea*, *Streblus*, and *Artocarpus*) in the Moraceae (*Nepal & Ferguson, 2012*). Recently, the genus *Morus* was redefined to contain eight species based on comprehensive analyses of ITS sequences from 43 mulberry accessions and one outgroup (*B. papyrifera*), and the phylogenetic relationships among clades were determined (*Zeng et al., 2015*). Most of the mulberry accessions collected in China were grouped into the largest clade B, even though there were many morphological polymorphisms and 12 supposed species among the mulberry accessions in clade B (*Zeng et al., 2015*). Because

the sequence similarity of ITS sequences in clade B was very high, it was difficult to redefine the classification and investigate the evolutionary patterns among them.

Generally, ITS regions are considered to be homologous in an individual as a result of concerted evolution (*Ghosh, Bhattacharya & Pal, 2017*; *Sang, Crawford & Stuessy, 1995*; *Wendel, Schnabel & Seelanan, 1995*), and intra-individual polymorphisms in ITS regions have been regarded as exceptions in many plant groups (*Mayol & Rossello, 2001*). As one type of ITS polymorphism, ITS pseudogenes were first found in *Zea mays* (*Buckler & Holtsford, 1996*) and then in various other plants (*Fan et al., 2014*; *Xiao, Moller & Zhu, 2010*). The ITS pseudogenes are easily distinguished by their GC content, minimum free energy of secondary structure, the presence of conserved motifs, substitution rates, phylogenetic positions, and copy numbers (*Bailey et al., 2003*; *Queiroz, Batista & De Oliveira, 2011*). It has been proposed that some ITS pseudogenes are inherited from the maternal progenitor, which is helpful for understanding the evolutionary history of a species (*Hughes, Bailey & Harris, 2002*). Putative ITS pseudogenes show higher sequence diversities than functional ITS sequences, and so they may provide better resolution and more information in phylogenetic analyses (*Xu et al., 2017*). In any case, ITS pseudogenes should be included in phylogenetic analyses (*Bailey et al., 2003*).

In preliminary analyses, we found ITS pseudogene sequences in mulberry, and speculated that they may provide new insights into mulberry evolution. Therefore, we conducted deep sequencing of ITS sequences from 33 mulberry accessions (seven species). We constructed phylogenetic trees using ITS sequences and the chloroplast DNA (cpDNA) sequences *trnL-trnF* and *trnT-trnL*, and found three connection patterns between ITS sequences and cpDNA sequences. Based on analyses of these patterns, we propose that hybridization has played important roles in the evolution of mulberry. The evolutionary history of mulberry is proposed. The results of this study contribute to our understanding of mulberry evolution.

## MATERIALS AND METHODS

### Plant materials

We selected 33 mulberry accessions (seven species: *M. alba*, *M. mongolica*, *M. cathayana*, *M. wittiorum*, *M. nigra*, *M. yunnanensis*, and *M. notabilis*) for this study (see Table 1). *Morus yunnanensis* was obtained from the Institute of Sericulture and Apiculture, Yunnan Academy of Agricultural Sciences, Mengzi, Yunnan province, China. The other mulberry accessions were obtained from the Mulberry Germplasm Nursery at Southwest University, China (*He et al., 2013*; *Zeng et al., 2015*), where they are preserved by propagation through grafting. The relatively closely related species *Artocarpus heterophyllus* was selected as the outgroup for the phylogenetic analyses based on ITS sequence (KT002551) and cpDNA sequences (MG434693).

### DNA preparation, sequence selection, and amplification

Genomic DNA was extracted from all mulberry accessions using the CTAB method (*Saghai-Maroof et al., 1984*). Chloroplast DNA was extracted as described by *Shi et al. (2012)*. Based on the alignment of six whole chloroplast genomes (from *M. notabilis*

**Table 1 Mulberry accessions used in this study and sequence characteristics of ITS, *trnL-trnF*, and *trnT-trnL*.**

| No. | Accessions | Taxa | Ploidy levels | ITS length (bp)# | ITS heterozygosity | Accession No. (ITS) | Clone number | *trnL-trnF* (bp) | Accession No. | *trnT-trnL* (bp) | Accession No. |
|---|---|---|---|---|---|---|---|---|---|---|---|
| 1 | Agentingsang | *M. alba* | 4x | 611/624 | 40% | MN044824/MN044849 | 20 | 921 | MN057958 | 1,096 | MN057991 |
| 2 | Banqiao6 | *M. alba* | 4x | 611 | 0% | MN044817 | 7 | 921 | MN057959 | 1,078 | MN057992 |
| 3 | Basailuona | *M. alba* | 4x | 611 | 0% | MN044831 | 7 | 921 | MN057961 | 1,078 | MN057994 |
| 4 | Gailiang10 | *M. alba* | 4x | 611 | 0% | MN044828 | 7 | 921 | MN057975 | 1,089 | MN057808 |
| 5 | Hanguodabaizhenzhu | *M. alba* | 4x | 611/611 | 14.29% | MN044825/MN044851 | 7 | 921 | MN057965 | 1,118 | MN057998 |
| 6 | Huasang | *M. alba* | 4x | 611 | 0% | MN044813 | 7 | 921 | MN057967 | 1,078 | MN057800 |
| 7 | Huai302 | *M. alba* | 12x | 611 | 0% | MN044836 | 7 | 921 | MN057966 | 1,078 | MN057999 |
| 8 | Huosang | *M. alba* | 4x | 611/624 | 5.26% | MN044833/MN044850 | 19 | 921 | MN057968 | 1,078 | MN057801 |
| 9 | Jianpuzhai | *M. alba* | 6x | 611/625 | 10% | MN044834/MN044854 | 10 | 921 | MN057969 | 1,112 | MN057802 |
| 10 | Leshandahongpi | *M. alba* | 6x | 611 | 0% | MN044823 | 7 | 921 | MN057971 | 1,111 | MN057804 |
| 11 | Lunjiao109 | *M. alba* | 4x | 611/624 | 25.00% | MN044822/MN044847 | 16 | 921 | MN057972 | 1,118 | MN057805 |
| 12 | Shanxitiansang | *M. alba* | 4x | 611/624 | 5% | MN044832/MN044846 | 20 | 921 | MN057976 | 1,079 | MN057809 |
| 13 | Shimiansang | *M. alba* | 4x | 611/611 | 14.29% | MN044818/MN044845 | 7 | 922 | MN057977 | 1,079 | MN057810 |
| 14 | Shuisang | *M. alba* | 4x | 611/611 | 14.29% | MN044821/MN044835 | 7 | 921 | MN057978 | 1,117 | MN057811 |
| 15 | Sililanka | *M. alba* | 4x | 611 | 0% | MN044830 | 7 | 921 | MN057980 | 1,078 | MN057813 |
| 16 | Taiwanchaochangguo | *M. alba* | 4x | 611 | 0% | MN044820 | 7 | 921 | MN057981 | 1,119 | MN057814 |
| 17 | Wupisang | *M. alba* | 4x | 611 | 0% | MN044814 | 7 | 922 | MN057982 | 1,078 | MN057815 |
| 18 | Xinjiaposijiguosang | *M. alba* | 4x | 611 | 0% | MN044827 | 7 | 921 | MN057979 | 1,118 | MN057812 |
| 19 | Xinyizhilai | *M. alba* | 4x | 611/624 | 20% | MN044829/MN044848 | 20 | 921 | MN057983 | 1,078 | MN057816 |
| 20 | Yidachimu | *M. alba* | 4x | 611 | 0% | MN044826 | 7 | 921 | MN057985 | 1,077 | MN057818 |
| 21 | Zhenzhubai | *M. alba* | 4x | 611 | 0% | MN044819 | 7 | 922 | MN057988 | 1,118 | MN057821 |
| 22 | Baojing7 | *M. cathayana* | 12x | 611/626 | 5% | MN044838/MN044852 | 20 | 922 | MN057960 | 1,112 | MN057993 |
| 23 | Gui23 | *M. cathayana* | 12x | 611 | 0% | MN044837 | 7 | 921 | MN057964 | 1,112 | MN057997 |
| 24 | Pisang2 | *M. cathayana* | 18x | 611/611 | 14.29% | MN044839/MN044844 | 7 | 921 | MN057974 | 1,112 | MN057807 |
| 25 | Jimengsang | *M. mongolica* | 4x | 611 | 0% | MN044815 | 7 | 921 | MN057970 | 1,078 | MN057803 |
| 26 | Mengsang | *M. mongolica* | 4x | 611 | 0% | MN044816 | 7 | 922 | MN057973 | 1,078 | MN057806 |
| 27 | *M. nigra* | *M. nigra* | 44x | 624 | 0% | KF784875 | 7 | 922 | MN057984 | 1,105 | MN057817 |
| 28 | *M. notabilis* | *M. notabilis* | 2x | 631 | 0% | KF784877 | 7 | 918 | MN057962 | 1,092 | MN057995 |
| 29 | Yun6 | *M. wittiorum* | 5x | 611/625 | 45% | MN044840/MN044856 | 20 | 921 | MN057986 | 1,112 | MN057819 |
| 30 | Yun6muben | *M. wittiorum* | 4x | 625/624 | 14.29% | MN044855/MN044843 | 7 | 921 | MN057989 | 1,112 | MN057822 |
| 31 | Yun7 | *M. wittiorum* | 7x | 611/626 | 5% | MN044842/MN044853 | 20 | 921 | MN057987 | 1,098 | MN057820 |
| 32 | Yun7muben | *M. wittiorum* | 8x | 611 | 0% | MN044841 | 7 | 921 | MN057990 | 1,098 | MN057823 |
| 33 | *M. yunnanensis* | *M. yunnanensis* | 2x | 631 | 0% | KF850474 | 7 | 918 | MN057963 | 1,094 | MN057996 |

**Note:**
#: ITS-α, α1 and Yun6muben-β1 are showed ahead in the column of ITS length.

(KP939360.1), *M. indica* (DQ226511.1), *M. cathayana* (KU981118.1), *M. mongolica* (KM491711.2), *M. alba* var. multicaulis (KM491711.2), and *M. alba* var. atropurpurea (KU355276.1)) using VISTA viewer (http://genome.lbl.gov/vista/index.shtml) (*Frazer et al., 2004*), we selected the two variable cpDNA sequences, *trnL-trnF* and *trnT-trnL*, for analyses. The ITS and the *trnL-trnF* sequences were amplified using primers described elsewhere (*Taberlet et al., 1991*; *Zeng et al., 2015*). We designed primers to amplify *trnT-trnL* (F: 5′-TGCGATGCTCTAACCTCT-3′; R: 5′-TAGCGTCTACCAATTTCG-3′). These sequences were amplified using GoTaq Flexi DNA Polymerase (Promega Corporation, Madison, WI, USA) according to the manufacturer's instructions. All the polymerase chain reaction (PCR) cycles consisted of initial denaturation of 95 °C for 5 min; followed by 32 cycles of 95 °C for 30 s, annealing for 30 s, 72 °C for 1 min; and then final extension for 7 min. The amplified fragments were isolated by electrophoresis on 1% (w/v) agarose gels. The purified sequences were cloned into the pMD19-T vector, and then 7–20 positive clones were sequenced for each accession.

## Sequence analyses

The cpDNA and ITS sequences were assembled and corrected for PCR errors using Sequencher 4.2 (Gene Codes Corp., Ann Arbor, MI, USA). All the sequences were aligned using Clustal X 1.81 software (*Thompson et al., 1997*). The boundaries of the ITS1, 5.8S, and ITS2 regions were determined as described elsewhere (*Zeng et al., 2015*). The GC content and sequence length was calculated using BioEdit (*Hall, 1999*). Three angiosperm conserved motifs (motif 1: GAATTGCAGAATCC, motif 2: TTTGAACGCA, motif 3: CGATGAAGAACGTAGC) were detected by BioEdit (*Kerbs et al., 2017*; *Yakimowski & Rieseberg, 2014*). The minimum free energy of the secondary structure was predicted using Mfold (http://www.bioinfo.rpi.edu/applications/mfold) (*Zuker, 2003*).

## Phylogenetic analyses

Phylogenetic analyses of ITS sequences and cpDNA sequences were conducted using maximum-likelihood (M-L) and Bayesian inference (BI) methods. The first phylogenetic analysis was conducted using the ITS sequences of the 33 mulberry accessions determined in this study and those from another nine species (*M. mesozygia* (HM747171), *M. insignis* (HM747169), *M. serrata* (HM747176), *M. rubra* (HQ144180), *M. celtidifolia* (HM747168), *M. macroura* (HM747170), *M. mongolica* (KF784879), *M. wittiorum* (AY345154), and *M. australis* (KT002555)) reported previously (*Zeng et al., 2015*). The best-fit model SYM+G for BI analyses of ITS sequences was selected by the lowest Akaike Information Criterion (AIC) scores in MrModelTest 2.3 (*Nylander et al., 2004*). MrBayes v3.2.6 software was used for BI analyses (*Huelsenbeck & Ronquist, 2001*). Four Markov chain Monte Carlo chains were run for 2,000,000 generations, with sampling every 100 generations. The first 5,000 trees were discarded as burn-ins and the 50% majority-rule consensus tree was determined to calculate the posterior probabilities for each node. The standard deviations of split frequencies were checked and the number of minimum generations required for analysis were those with a standard

deviation value lower than 0.01. The M-L phylogenetic trees were constructed using the most suitable Kimura 2-parameter model with the lowest Bayesian Information Criterion scores in MEGA 7 (*Kumar, Stecher & Tamura, 2016*). A discrete gamma distribution was used to model evolutionary rate differences among sites (five categories (+G, parameter = 1.3541)) (*Kumar, Stecher & Tamura, 2016*). All characters were equally weighted.

Phylogenetic analyses of cpDNA sequences were performed using sequence matrix data for the *trnL-trnF* and *trnT-trnL* regions. The best-fit model (GTR) for BI and M-L analyses for cpDNA sequences was selected by the lowest AIC scores in MrmodelTest 2.3 (*Nylander et al., 2004*). The other parameters for BI analyses based on cpDNA sequences were the same as the parameters for analyses based on ITS sequences. The M-L phylogenetic tree was constructed in MEGA 7 with 1,000 bootstrap replicates (*Kumar, Stecher & Tamura, 2016*). All characters were equally weighted.

# RESULTS

## Variations among ITS sequences

Thirty-three mulberry accessions were sequenced and their ITS sequences were analyzed (Table 1). For each accession, 7–20 clones were sequenced (Table 1). Of the 33 mulberry accessions, 19 contained only one type of ITS sequence, with lengths ranging from 611 bp to 631 bp. The ITS sequences of *M. notabilis*, *M. yunnanensis*, and *M. nigra* were confirmed to have lengths of 631 bp, 631 bp, and 624 bp, respectively, as reported previously (*Zeng et al., 2015*). Of the 19 mulberry accessions with one ITS sequence, 16 had an ITS sequence of 611 bp, with differences at only two single nucleotide polymorphism (SNP) sites.

The other 14 mulberry accessions had polymorphic ITS sequences (Table 1). There were two types of ITS sequences: short ITS sequences (ITS-α) and long ITS sequences (ITS-β). The ITS sequences of the mulberry accessions Shimiansang, Hanguodabaizhenzhu, Shuisang, and Pisang2 contained two short-type sequences. Sequences with higher and lower copy numbers were designated as ITS-α1 and ITS-α2, respectively. The accession Yun6muben contained two long ITS sequences (625 bp and 624 bp, designated as ITS-β1 and ITS-β2, respectively). The ITS-α sequences in 13 mulberry accessions were identical with a length of 611 bp. The length of ITS-β sequences ranged from 624 bp to 626 bp. An alignment of the ITS sequences of the mulberry accessions is shown in Fig. 1. We detected the 13-bp InDel reported previously (*Zeng et al., 2015*). Further, we found three types of 13-bp sequences among the 33 mulberry accessions (CGTATACAATGCG, TGTGTGCAATGCG, and CGTACACAATGCG). Alignment analyses of the ITS sequences revealed six 1-bp and three 2-bp InDels. Other sequence variations were SNPs.

## Identification of ITS pseudogenes

The GC content of the ITS region, the minimum free energy of the secondary structure, and conserved motifs in the 5.8S rDNA region were used to identify ITS pseudogenes. As shown in Fig. 2 and Table S1, the GC contents of Shuisang-α2, Jianpuzhai-β, and

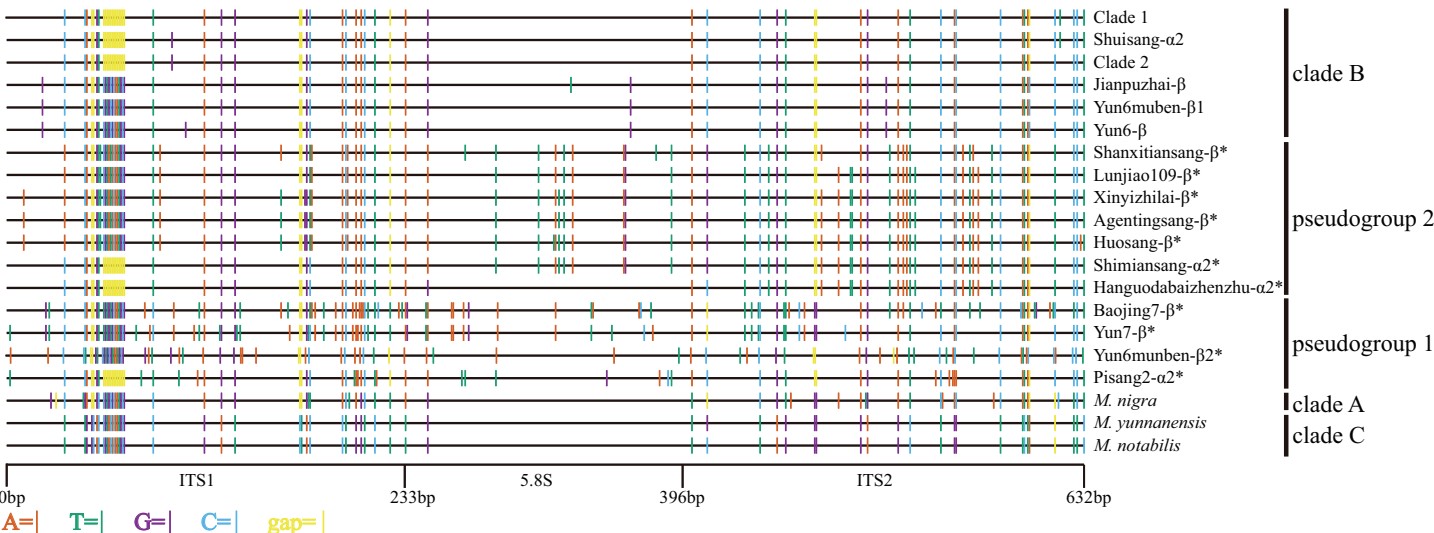

**Figure 1 Distribution of different sites across ITS region.** ITS sequence of *Morus notabilis* was used as reference. Colored lines indicate sites with lower ratios: vermilion line, A; bluish green line, T; purple line, G; sky blue line, C; yellow line, gap; black line, whole ITS sequence. ITS pseudogene sequences were marked with *. The clade names were come from Fig. 3.

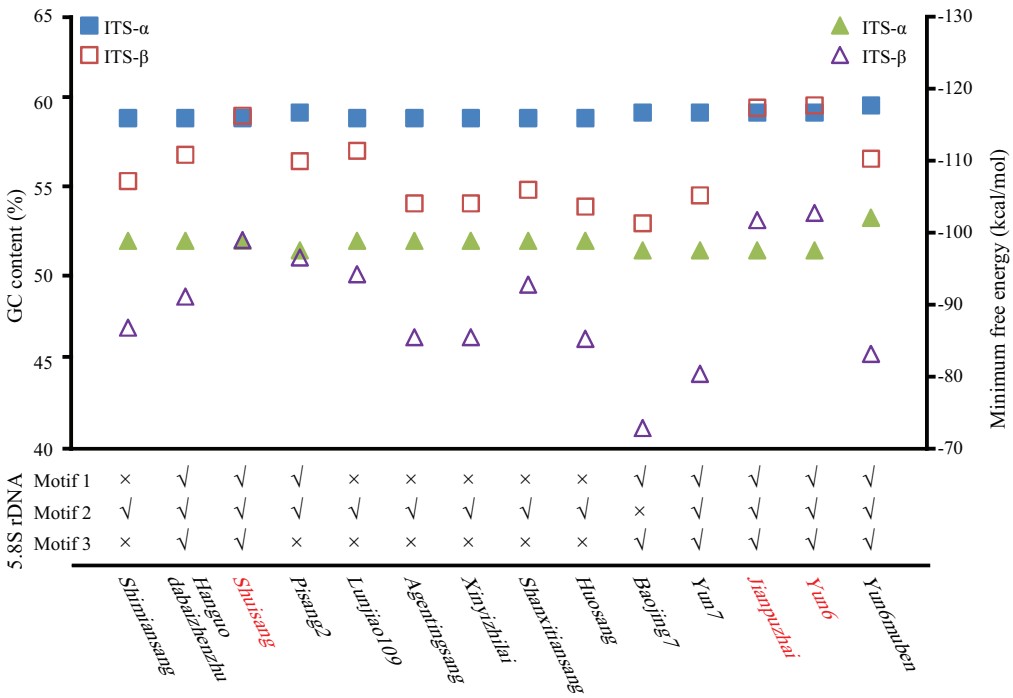

**Figure 2 Identification of ITS pseudogenes.** GC content (quadrangle), minimum free energy (triangle), and conserved motifs in 5.8S rDNA were analyzed. Shuisang, Jianpuzhai, and Yun6 have two functional ITS sequences and are shown in red.

Yun6-β were similar to that of ITS-α1/α. The GC content of Yun6muben-β2 was lower than that of ITS-β1. The GC contents of ITS-α2/β2 sequences in the remaining mulberry accessions were lower than those of ITS-α1/β1 sequences in at least one region of ITS1, 5.8S rDNA, and ITS2. The minimum free energy of secondary structure showed a similar

pattern to that of GC contents, except that it was lower in Shuisang-α2, Jianpuzhai-β, Yun6-β, and Yun6muben-β2 than in ITS-α1/α and Yun6muben-β1. The sequences of Hanguodabaizhenzhu-α2, Shuisang-α2, Yun7-β, Jianpuzhai-β, Yun6-β, and Yun6muben-β2 contained all three conserved motifs. Based on the sequence length information, the GC content of the ITS region, the minimum free energy of secondary structure, and conserved motifs in the 5.8S rDNA region, we identified 11 ITS sequences as pseudogenes (Yun6muben-β2, Pisang2-α2, Shimiansang-α2, Shanxitiansang-β, Xinyizhilai-β, Lunjiao109-β, Huosang-β, Agentingsang-β, Hanguodabaizhenzhu-α2, Baojing7-β, and Yun7-β).

### ITS phylogenetic analyses

We included *M. mesozygia* (clade D1), *M. insignis* (clade D2), *M. serrata* (clade A4), *M. rubra* (clade A3), *M. celtidifolia* (clade A1), *M. macroura* (clade B), *M. mongolica* (clade B), *M. wittiorum* (clade B), *and M. australis* (clade B) in the phylogenetic analyses, as they represented six of the eight clades classified in a previous study (*Zeng et al., 2015*). We constructed M-L and BI trees. The main clades in the M-L trees (Fig. 3) were the same as those in the BI trees (Fig. S2), and both were consistent with the results of a previous study (*Zeng et al., 2015*). However, the pseudogene Yun6muben-β2 clustered together with the functional ITS sequences in the BI tree (Fig. S2), so we used the M-L tree for further analyses. Most of the mulberry accessions were clustered in clade B in these two trees. In clade B, all the functional ITS sequences were grouped into three clades (clade 1, clade 2, and clade 3) (Fig. 3): two short-type ITS sequence clades (clade 1 and clade 2) and a long-type ITS sequence clade (clade 3). The ITS pseudogenes were grouped into two pseudogene clades, designated as pseudogroup 1 and pseudogroup 2 (Fig. 3).

### Characteristics and phylogenetic analyses of cpDNA sequences

Based on the synteny analyses of the whole chloroplast genomes of *M. indica* (*Ravi et al., 2007*), *M. notabilis* (*Chen et al., 2016a*), *M. mongolica* (*Kong & Yang, 2016*), *M. cathayana* (*Kong & Yang, 2017*), *M. alba* var. multicaulis (*Kong & Yang, 2017*), and *M. alba* var. atropurpurea (*Hu et al., 2014*) (Fig. S1) and sequence alignment analyses, the *trnL-trnF* and *trnT-trnL* regions were selected for the cpDNA phylogenetic analyses. In the 33 mulberry accessions, the length of *trnL-trnF* ranged from 913 bp to 931 bp, and that of *trnT-trnL* ranged from 1108 bp to 1150 bp. The combined data matrix of *trnL-trnF* and *trnT-trnL* consisted of 2109 aligned nucleotides (Fig. 4).

We constructed M-L and BI phylogenetic trees using the *trnL-trnF* and *trnT-trnL* data (Fig. 5; Fig. S3). The topologies of the BI trees were disordered (Fig. S3), and some identical sequences were clustered into different clades. Therefore, the M-L trees were used for further analyses. First, *M. notabilis* and *M. yunnanensis* were diverged first among the 33 mulberry accessions. The remaining 31 mulberry accessions were separated into two main clades. Clade I contained 11 accessions of species *M. alba* (Banqiao6, Basailuona, Huasang, Huai302, Huosang, Shanxitiansang, Shimiansang, Sililanka, Xinyizhilai,

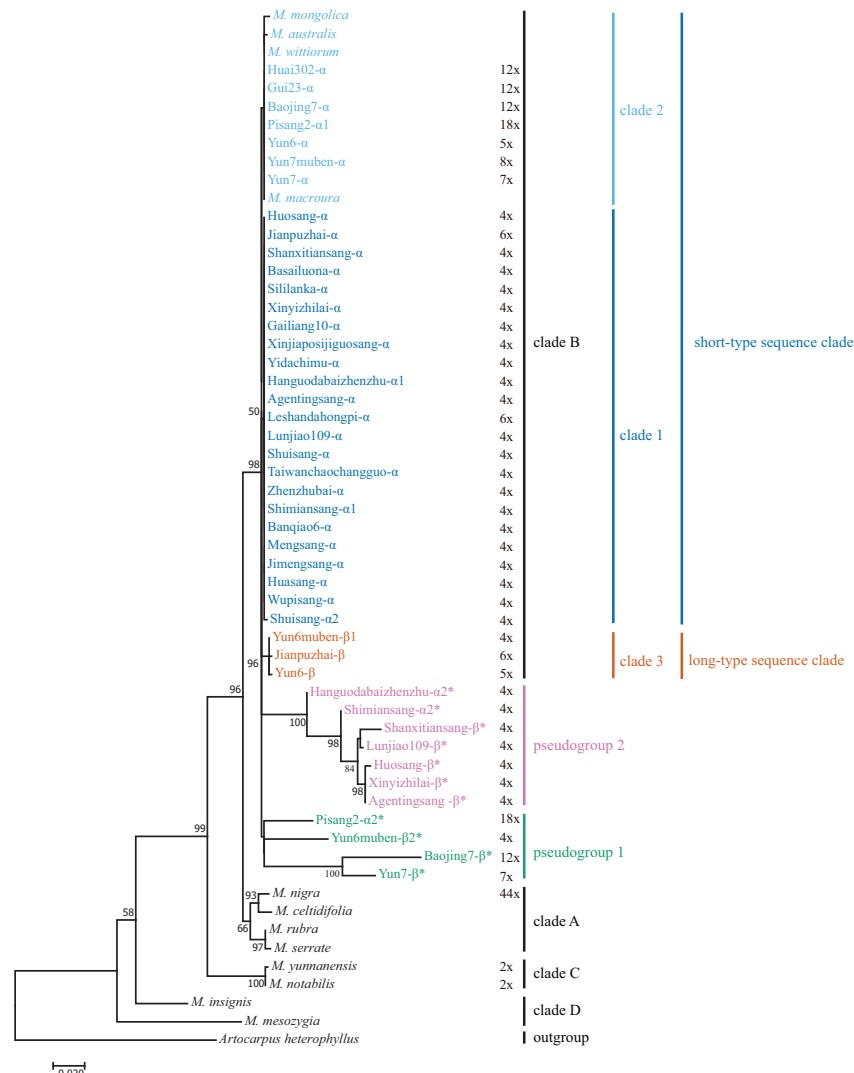

**Figure 3 Maximum-Likelihood phylogenetic tree based on ITS sequences of 33 mulberry accessions.**
Phylogenetic tree was constructed using Kimura 2-parameter model in MEGA 7. Clade B was separated
into short-type sequence clade (blue clade 1 and sky blue clade 2), long-type sequence clade (vermilion
clade 3, bluish green pseudogroup 1, and reddish purple pseudogroup 2). Other clades reported by
(Zeng et al., 2015) were shown in black font. ITS pseudogene sequences were marked with *. Bootstrap
support values for ML below 50% are not shown.             

Yidachimu, and Wupisang), two accessions of species *M.mongolica* (Jimengsang and
Mengsang) (Figs. 4 and 5; Table 1). Clade II contained 10 accessions of species *M. alba*
(Agentingsang, Gailiang10, Hanguodabaizhenzhu, Jianpuzhai, Leshandahongpi,
Lunjiao109, Shuisang, Taiwanchaochangguo, Xinjiaposijiguosang, and Zhenzhubai), three
accessions of species *M. cathayana* (Baojing7, Gui23, and Pisang2), four accessions of
species *M. wittiorum* (Yun6, Yun6muben, Yun7, and Yun7muben), and one accession of
species *M. nigra* (Fig. 5; Table 1). There were more sequence variations in clade II than in
clade I (Fig. 4). Species of *M. alba* (Hanguodabaizhenzhu, Lunjiao109, Shuisang,

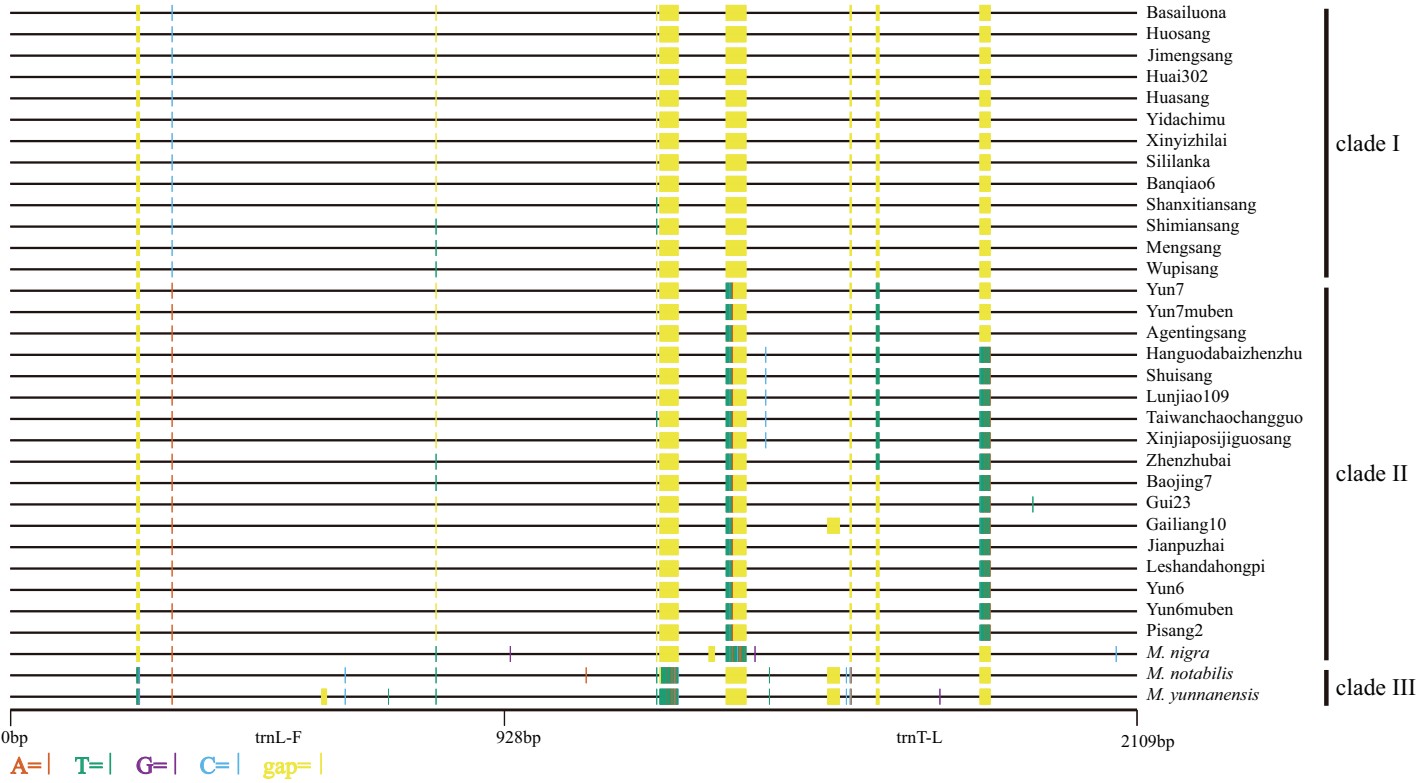

**Figure 4 Distribution of different sites across *trnL-trnF* and *trnT-trnL* regions.** Sequence of *Morus notabilis* was used as reference. Colored sites indicate sites with lower ratios: vermilion line, A; bluish green line, T; purple line, G; sky blue line, C; yellow line, gap; black line, whole *trnL-trnF* and *trnT-trnL* sequences. The clade names were come from Fig. 5.

Taiwanchaochangguo, and Xinjiaposijiguosang) formed a subclade, and *M. nigra* was separated from the other mulberry accessions on a long branch (Fig. 5).

## DISCUSSION

### Characterization and mutation pattern of ITS pseudogenes

ITS pseudogenes have been detected in many plants (*Fan et al., 2014*; *Won & Renner, 2005*; *Xiao, Moller & Zhu, 2010*; *Xu et al., 2017*), and should not be excluded from phylogenetic analyses of ITS sequences (*Bailey et al., 2003*). However, ITS pseudogenes have evolved at a faster rate than functional ITS sequences, which can cause confusion in phylogenetic analyses (*Bailey et al., 2003*; *Fan et al., 2014*). Thus, comprehensive analyses of ITS sequences are required. The ITS pseudogenes can be easily identified based on their GC content, minimum free energy of secondary structure, presence of conserved motifs, substitution rates, phylogenetic positions, and copy numbers (*Bailey et al., 2003*; *Queiroz, Batista & De Oliveira, 2011*). In the present study, we found ITS pseudogenes in 11 mulberry accessions, implying that incomplete concerted evolution is occurring in mulberry. Compared with functional ITS sequences, the ITS pseudogenes of mulberry showed higher diversity in the whole ITS region (ITS1, 5.8S rDNA, and ITS2), as found in other studies (*Fan et al., 2014*; *Xiao, Moller & Zhu, 2010*). Both the

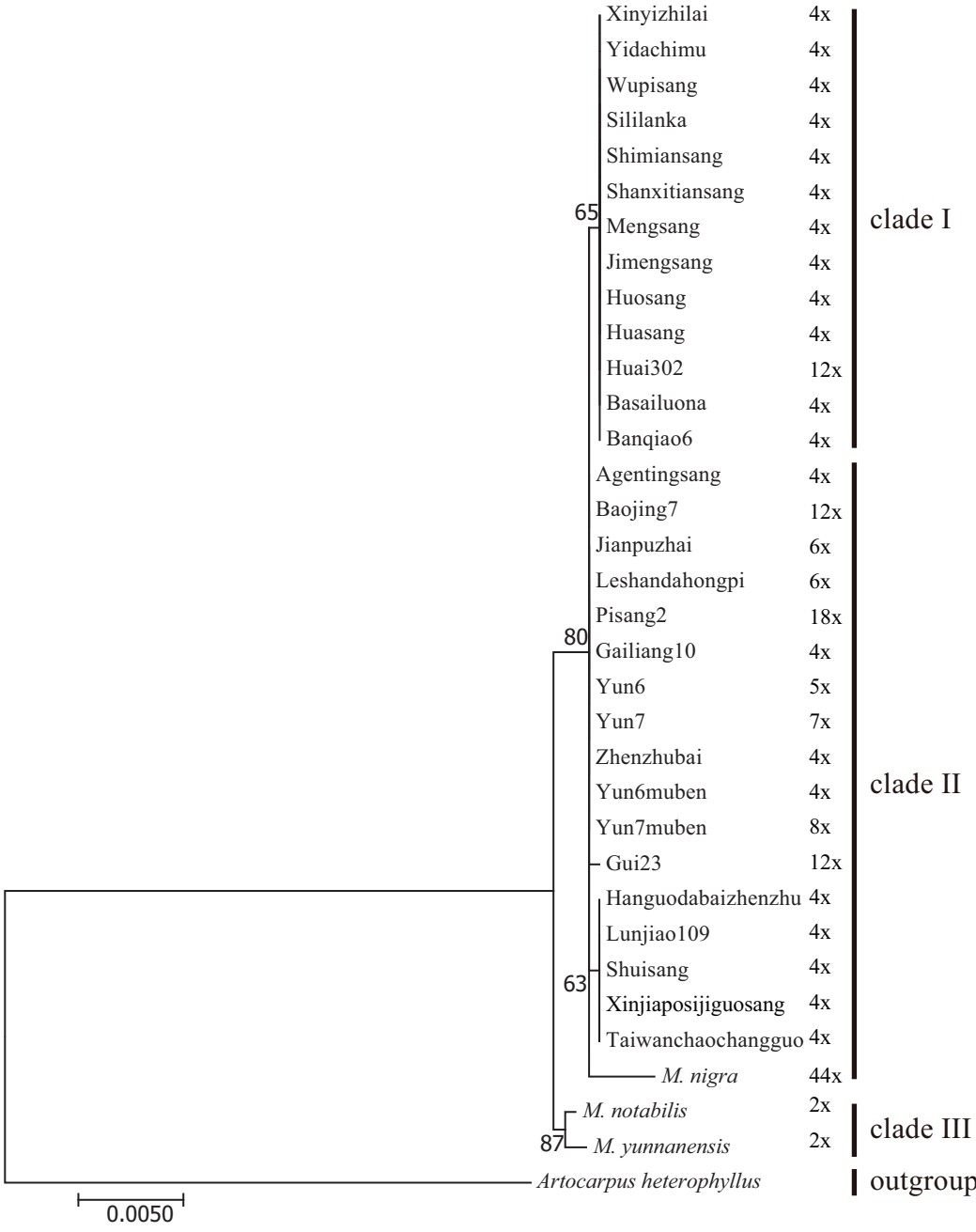

| | | |
|---|---|---|
| Xinyizhilai | 4x | |
| Yidachimu | 4x | |
| Wupisang | 4x | |
| Sililanka | 4x | |
| Shimiansang | 4x | |
| Shanxitiansang | 4x | clade I |
| Mengsang | 4x | |
| Jimengsang | 4x | |
| Huosang | 4x | |
| Huasang | 4x | |
| Huai302 | 12x | |
| Basailuona | 4x | |
| Banqiao6 | 4x | |
| Agentingsang | 4x | |
| Baojing7 | 12x | |
| Jianpuzhai | 6x | |
| Leshandahongpi | 6x | |
| Pisang2 | 18x | |
| Gailiang10 | 4x | |
| Yun6 | 5x | |
| Yun7 | 7x | |
| Zhenzhubai | 4x | clade II |
| Yun6muben | 4x | |
| Yun7muben | 8x | |
| Gui23 | 12x | |
| Hanguodabaizhenzhu | 4x | |
| Lunjiao109 | 4x | |
| Shuisang | 4x | |
| Xinjiaposijiguosang | 4x | |
| Taiwanchaochangguo | 4x | |
| *M. nigra* | 44x | |
| *M. notabilis* | 2x | clade III |
| *M. yunnanensis* | 2x | |
| *Artocarpus heterophyllus* | | outgroup |

0.0050

**Figure 5 Maximum-likelihood phylogenetic tree based on *trnL-trnF* and *trnT-trnL* regions of 33 mulberry accessions.** Phylogenetic tree was constructed using Kimura 3-parameter model in MEGA 7; 33 mulberry accessions clustered into clade I, clade II, and clade III.

pseudogenes and functional ITS sequences were a mixture of short- and long-type sequences. Recombination was detected by the RDP4 program (*Martin et al., 2015*). The results showed that the short-type pseudogene sequences Shimiansang-α2 and Hanguodabaizhenzhu-α2 have recombined with putative long-type ITS sequences, representing one pathway of concerted evolution of ITS sequences.

## Phylogenetic analyses and evolution of ITS

We conducted phylogenetic analyses of ITS functional sequences and pseudogenes. The main topologies of the phylogenetic trees were the same as that reported elsewhere (*Zeng et al., 2015*). In the present study, ITS pseudogenes were identified and separated from the functional ITS clades (Fig. 3) (*Bailey et al., 2003*). Clade 3 is a newly identified clade with long-type ITS sequences compared with the previous study (*Zeng et al., 2015*). Based on the sequence similarities and sequence length of the ITS region, the long-type ITS sequences in clade 3 may have closer relationships with the sequences in clades A, C, and D (Fig. 1).

In our analyses, most of the sequences were clustered in clade 1 and clade 2, with only three sequences clustered into clade 3 (Fig. 3). We detected inconsistencies between traditional and molecular classifications in this study. Twenty-nine mulberry accessions with morphological polymorphisms clustered together with a low phylogenetic resolution and had similar or identical ITS sequences. For example, Yun6muben and Yun7muben showed similar fruit, leaf, bud, and internode morphologies, indicating that both should be classified as *M. wittiorum*. However, Yun6muben had long-type ITS sequences that clustered in clade 3, while Yun7muben had short-type ITS sequences that clustered in clade 2. As the morphological characteristics were very complex, no details are discussed here. These phenomena are indicative of concerted evolution or introgression of the ITS region, and of hybridization (*Bailey et al., 2003*; *Xu et al., 2017*). Thus, the classification of mulberry is very complex, and there is considerable confusion at present. Systematic classification of mulberry resources should be based on molecular markers, morphology, chromosome number, genome data, and other characteristics.

Intra-individual polymorphisms in the ITS region have been detected in many plants and pseudogene sequences have been included in phylogenetic analyses (*Mayol & Rossello, 2001*; *Xiao, Moller & Zhu, 2010*; *Zheng et al., 2008*). Bidirectional, unidirectional, or different rates of evolution have been detected from analyses of ITS sequences (*Wendel, Schnabel & Seelanan, 1995*; *Xu et al., 2017*; *Zheng et al., 2008*). In this study, we detected multiple functional ITS copies, putative ITS pseudogenes, or recombinants of ITS pseudogenes in the same individual. The successive evolution patterns of ITS sequences are indicative of concerted evolution (Figs. 1 and 3). There are several lines of evidence for the concerted evolution of ITS sequences. First, clades A, C, D, and 3 had long-type and ancient ITS sequences. Second, Yun6 was identified as the progeny of Yun6muben through natural pollination. Yun6-$\beta$ showed one SNP variation compared with Yun6muben-$\beta 1$, suggesting that concerted evolution started in the F1 generation. Third, Hanguodabaizhenzhu-$\alpha 2$ and Shimiansang-$\alpha 2$ were found to be recombined from long-type ITS sequences, and other ITS pseudogenes showed more variations. Fourth, Shuisang-$\alpha 2$ had only one SNP variation compared with ITS sequences in clade 1 and clade 2, implying that concerted evolution is continuing or has completed only recently. Finally, most functional ITS copies were short-type sequences. In conclusion, the evolutionary process of ITS, as indicated by our data, is that long-type ITS sequences are undergoing concerted evolution to form short-type ITS sequences.

Hybridization is an important evolutionary mechanism in plants, especially in flowering plants (*Kerbs et al., 2017*; *Soltis & Soltis, 2009*; *Yakimowski & Rieseberg, 2014*). Hybridization can also help to explain the concerted evolution of ITS sequences (*Xu et al., 2017*). We detected three hybridized mulberry accessions (Jianpuzhai, Shuisang, and Yun6), which contained two functional ITS sequences. Hybridization between *M. rubra* (A clade) and *M. alba* (B clade) has been detected in previous studies (*Burgess & Husband, 2004*; *Burgess, Morgan & Husband, 2008*). It has been suggested that hybridization between *M. rubra* and *M. alba* caused the local decline of *M. rubra* (*Burgess et al., 2005*). This may indicate that *M. alba* has higher ecological potential. The continuous back-cross hybridization of *M. rubra* with *M. alba* may have led to the disappearance of ITS sequences from *M. rubra*, or the concerted evolution to form short-type sequences. Like other plants, local *Morus* species face the risk of extinction (*Burgess & Husband, 2004*; *Burgess, Morgan & Husband, 2008*; *Ellstrand & Schierenbeck, 2000*; *Wolf, Takebayashi & Rieseberg, 2001*).

Polyploidy is another important evolutionary mechanism that is known to affect the concerted evolution of ITS sequences in plants (*Ainouche et al., 2004*). The ITS sequences can retain their subgenomic sequences or become homogenized through concerted evolution (*Wendel, Schnabel & Seelanan, 1995*). Existing mulberry species show various polyploidy levels, with 14, 28, 35, 42, 49, 56, 84, 112, 126, or 308 chromosomes (*Xuan et al., 2017*; *Zeng et al., 2015*). Multiple nucleolus organizing region (NOR) loci have also contributed to the concerted evolution of ITS sequences. Two pairs of NOR loci have been reported for *M. notabilis* (*Xuan et al., 2017*). The other mulberry accessions contained at least two pairs of NOR loci (data not shown). Mulberry is often cultivated by grafting or propagated from cuttings, resulting in a long generation time. This may be another factor affecting the concerted evolution of ITS sequences. In summary, hybridization, polyploidy, multiple NOR loci, and long generation times have all contributed to the concerted evolution of ITS sequences in mulberry.

## Phylogenetic analyses of cpDNA sequences

Certain cpDNA sequences are widely used molecular markers in phylogenetic analyses (*Nepal & Ferguson, 2012*; *Xu et al., 2012*). With the development of universal primers for cpDNA and chloroplast genome sequencing, increasing numbers of studies have been conducted based on cpDNA (*Huang et al., 2014*; *Wu et al., 2014*). Another characteristic of cpDNA is that it is maternally inherited. To date, six chloroplast genomes of mulberry have been reported, and they can provide comprehensive information about the evolution of the whole chloroplast genome (*Chen et al., 2016a*; *Hu et al., 2014*; *Kong & Yang, 2016*, *2017*; *Ravi et al., 2007*). Several other molecular markers have been used in phylogenetic studies, with *trnL-trnF* being the most commonly used (*Ayinampudi et al., 2011*; *Nepal & Ferguson, 2012*). In this study, more variable cpDNA regions (*trnL-trnF* and *trnT-trnL*) were used for phylogenetic analyses to study the concerted evolution of mulberry. The mulberry accessions in clade B were separated into two clades in the phylogenetic tree based on cpDNA sequences. The sequences in clade I were almost identical, sequences in clade II showed more variations and a subclade was clustered (Figs. 4 and 5). Successive

evolution pattern of the sequences was shown in Fig. 4, suggesting a closer progenitor among them. Thus, clade II could be treated as a single clade. These results were consistent with those of previous studies in which two clades were proposed based on phylogenetic analyses of *TrnL-TrnF* (*Zhao et al., 2005*; *Zhao et al., 2007*).

## Evolution of mulberry

Based on the analyses of ITS sequences, the 33 functional ITS sequences in clade B were separated into short-type sequence clades (clade 1 and clade 2) and a long-type sequence clade (clade 3). Two pseudogroups were also divided in the ITS phylogenetic tree. These findings suggest that there is a shallow level of phylogeny among 29 mulberry accessions, consistent with the two clades detected in the cpDNA phylogenetic tree (*Bailey et al., 2003*). Thus, clade B contains two species.

On the basis of a study of the genus *Leucaena*, it was proposed that some ITS pseudogene sequences are inherited from the maternal progenitor (*Hughes, Bailey & Harris, 2002*). In this study, we detected three connection patterns between ITS pseudogene sequences and cpDNA sequences (Figs. 1 and 4). Connection pattern (1) was between pseudogroup 2 and clade I, and was detected in Shanxitiansang, Xinyizhilai, Huosang, and Shimiansang. In this pattern, both the ITS pseudogene sequences and cpDNA sequences showed high sequence similarities. Connection pattern (2) was between pseudogroup 1 and clade II, and was detected in Yun7, Yun6muben, Baojing7, and Pisang2. The ITS pseudogene sequences of these four mulberry accessions displayed sequence variations, but the cpDNA sequences did not. Connection pattern (3) was between pseudogroup 2 and clade II, and was detected in Hanguodabaizhenzhu, Lunjiao109, and Agentingsang. In these three patterns, the ITS pseudogene sequences have connections with genetic information from the female progenitor. This provides more clues to trace the evolution process of ITS sequences and the *Morus* genus.

The three connection patterns and the incongruent distribution patterns between the cpDNA and the ITS phylogenetic trees imply that hybridization has contributed to the evolution of mulberry. Based on the discussion above, a simple evolutionary pattern is proposed for mulberry (Fig. 6). First, mulberry with short-type ITS sequences ([a]AA) existed at a certain time and showed stronger ecological potential. Those resources then hybridized with mulberry with long-type sequences ([b]BB). The ITS sequences gradually became short-type sequences through concerted evolution or hybridization. Finally, long-type ITS sequences (B) were left as pseudogene sequences (B) in the offspring, and most mulberry had short-type functional ITS sequences ([a]AA, [a]AA[B], and [b]AA[B]). Thus, mulberry with long-type ITS sequences ([b]BB) are the original mulberry resources, and are being polluted by those with short-type ITS sequences during evolution. The actual evolutionary process of mulberry will be more complex than this evolutionary pattern, and dominated by hybridization.

In the phylogenetic trees constructed from ITS and cpDNA sequences (Figs. 3 and 5), all the clades contained accessions with different ploidy levels, and there were cross-links between the clades in different trees (e.g., Huai302 ($2n = 12x = 84$) belonged to clade 2 and clade I). These results indicate that hybridization and polyploidy have played important

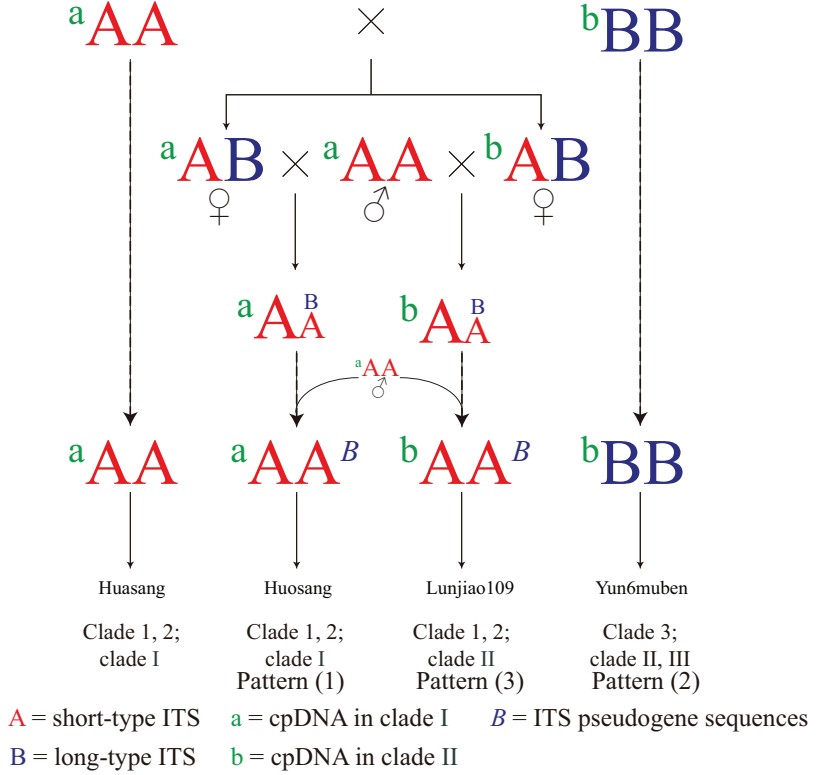

**Figure 6 Simple evolutionary pattern of mulberry.** Mulberry with short-type ($^a$AA) and long-type ($^b$BB) ITS sequences continuously hybridized with mulberry with short-type ITS sequence mulberry ($^a$AA). Finally, $^a$AA mulberry have become most abundant. Mulberry with $^a$AA$^B$ and $^b$AA$^B$ contain ITS pseudogene sequences ($B$) from different female progenitors. $^b$BB represents mulberry with long-type ITS sequences. Dotted arrow represents continuous hybridization process.

roles in the evolution of mulberry, even in the accessions with higher ploidy levels. Clade 2 of the phylogenetic tree constructed from ITS sequences contained mulberry accessions with different high ploidy levels, identical ITS sequences, and successively evolved cpDNA sequences, implying that concerted evolution has played a key role in the evolution of this clade. In conclusion, polyploidy, hybridization, and concerted evolution have all played important roles in the evolution of ITS sequences in mulberry.

## CONCLUSIONS

Based on phylogenetic analyses of ITS sequences and cpDNA sequences, clade B was separated into two species. We found ITS pseudogenes in mulberry, and detected a concerted evolutionary process in the direction of short-type ITS sequences. We detected three connection patterns between ITS pseudogene sequences and cpDNA sequences, suggesting that the ITS pseudogene sequences connect with genetic information from the female progenitor. Combining the three connection patterns and incongruent distribution patterns between phylogenetic trees constructed from cpDNA and ITS sequences, hybridization is recent or still occurring, and has played important roles in mulberry

evolution. The proposed evolutionary pattern, in which hybridization is a key feature, helps us to understand the evolutionary history of mulberry and highlights the importance of conserving local resources. Finally, polyploidy, hybridization, and concerted evolution have all played roles in the evolution of ITS sequences in mulberry.

### Funding

This project was funded by the National Key Research and Development Program (No. 2018YFD1000602), the Natural Science Foundation of China (No. 31572323), the special fund for Agro-scientific research in the public interest of China (No. 201403064), the "111" Project (B12006), the Chongqing Research Program of Basic Research and Frontier Technology (cstc2018jcyjAX0407), and the Fundamental Research Funds for the Central Universities (SWU118040). The funders had no role in study design, data collection and analysis, decision to publish, or preparation of the manuscript.

### Grant Disclosures

The following grant information was disclosed by the authors:
National Key Research and Development Program: 2018YFD1000602.
Natural Science Foundation of China: 31572323.
Agro-Scientific Research in the Public Interest of China: 201403064.
"111" Project: B12006.
Chongqing Research Program of Basic Research and Frontier Technology: cstc2018jcyjAX0407.
Fundamental Research Funds for the Central Universities: SWU118040.

### Competing Interests

The authors declare that they have no competing interests.

### Author Contributions

- Yahui Xuan conceived and designed the experiments, performed the experiments, analyzed the data, contributed reagents/materials/analysis tools, prepared figures and/or tables, authored or reviewed drafts of the paper, approved the final draft.
- Yue Wu performed the experiments, analyzed the data, contributed reagents/materials/analysis tools, prepared figures and/or tables, authored or reviewed drafts of the paper, approved the final draft.
- Peng Li performed the experiments, prepared figures and/or tables, approved the final draft.
- Ruiling Liu performed the experiments, approved the final draft.
- Yiwei Luo performed the experiments, contributed reagents/materials/analysis tools, prepared figures and/or tables, approved the final draft.
- Jianglian Yuan performed the experiments, contributed reagents/materials/analysis tools, prepared figures and/or tables, approved the final draft.

- Zhonghuai Xiang conceived and designed the experiments, authored or reviewed drafts of the paper, approved the final draft, suggested the manuscript.
- Ningjia He conceived and designed the experiments, analyzed the data, prepared figures and/or tables, authored or reviewed drafts of the paper, approved the final draft.

## DNA Deposition

The following information was supplied regarding the deposition of DNA sequences:

The raw data of ITS sequences and cpDNA sequences are available in Zenodo: DOI 10.5281/zenodo.3238365.

## Data Availability

The sequences are available at NCBI: ITS sequences, MN044813–MN044856; trnL-trnF sequences, MN057958–MN057990; trnT-trnL sequences, MN057991–MN058023.

## Supplemental Information

Supplemental information for this article can be found online at http://dx.doi.org/10.7717/peerj.8158#supplemental-information.

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
