# Peer review of "Molecular phylogeny of mulberries reconstructed from ITS and two cpDNA sequences"

_PeerJ, doi:10.7717/peerj.8158_

## Round 0.1 · original submission · Major Revisions

Please revise your manuscript to the reviewers' comments. Also, clearly state the additional findings compared to the paper "Zeng et al. (2015) Definition of Eight Mulberry Species in the Genus Morus by Internal Transcribed Spacer- Based Phylogeny. PLoS ONE 10(8): e0135411".

·

Basic reporting

The first word in the title is “Classification” but there is no clear presentation of any taxonomy based on the results. Results are only communicable via scientific names and this has not been done in any clear way. The authors show no knowledge of standard taxonomic practices when presenting their results, they would do well to find a taxonomist to be a collaborator. As it is, it is quite difficult to find out the actual taxonomically significant results, particularly how to apply them without access to a molecular laboratory.

Experimental design

Need to clarify what is meant by ‘sample’, are these single living trees or groups of trees? what are the provenances of these ‘samples’? are there permanently preserved vouchers? The authors state that 7-20 clones sequenced from each sample, are these from different trees or different parts of a single tree? were these vegetatively propagated?

Validity of the findings

The conclusions are totally inadequate as there is no clear link to the actual classification of Morus.

Additional comments

line 72: consistent with the morphological classification, not ‘consisting’ with …
line 75: Nepal et al. recognised 13 species, not … ‘classified’ 13 species (or accepted 13 …)
Nepal et al. provided evidence that Morus is paraphyletic because two species of Trophis were embedded within it in their analysis but there is no follow up on this even though this account is supposed to be a review of the genus.
line 182: this section needs to be rewritten so as to clarify the identities of the various clades - at present there is nothing connecting the initial list of taxa and the clades / groups, especially as it listed 9 taxa and then refers to seven out of eight clades.
line 246: there are no details of the morphological polymorphisms mentioned, these need to be described / tabulated in some detail and their possible origins discussed. There can be significant differences between juvenile and adult leaves, how relevant is this?
line 254: Is not the practical need to find which morphological features correlate best with the groupings based on the molecular data and chromosome numbers? That way the results can be used outside of a molecular lab.
line 298: what taxa correspond with these ‘molecular species’?
line 329: Fig. 6 needs to be linked to names, how does it relate to the M. alba / M. rubra hybridization discussed in the previous paragraph?
line 347: The sentence starting ‘However’ needs rewriting, mulberries are normally propagated vegetatively thus giving long generation times. ‘sticking’ is not a good translation (too ambiguous), better ‘propagated via cuttings’ perhaps “However, the cultivated examples of Mulberries have been propagated vegetatively via grafting and cuttings, giving long generation times, which may . . .”

Reviewer 2 ·

Basic reporting

The English used here are not very clear and I encountered many occasions where the presentation are ambiguous. Although the authors provide many literature references, I cannot guarantee that they have enough background on evolutionary biology, which is essential to this article. The article do not have a very good structure to broadcast their ideas to the readers, since they have not distilled their idea carefully and thoroughly. I also do not see a clear hypothesis that the authors have clearly presented.

Experimental design

Methods described with sufficient details and information to replicate. However, the research question may need to be further distilled.

Validity of the findings

All underlying data have been provided and they are robust.

Additional comments

Mulberry is one of the most improtant economic crop in ancient China, where the economic chain of silk road rest on. Doubtless, there are numerous times that the cultivated populations of mulberry in China were domesticated from wild populations, or be crossed between cultivated populations or between cultivated and wild populations. In this manuscript, the authors did a good job to infer hybridization between different evolutionary lineages in the light of ITS sequences, pseudogenes of ITS sequences and cpDNA sequences from 33 samples in China, in the help that cpDNA and pseudogenes of ITS sequences are inherited form maternal progenitor. However, I found that the manuscript await substantial revision before being considered for publication.

First of all, I think the author may need to distill the key questions they want to address in this manuscript, and then restructure the introduction, as well as the discussion section. Although I found Fig. 6 very attractive, it seems that the authors do not tell a story on this topic, they start their manuscript with introducing of ITS! Mulberry and their evolutionary history are much more attracting! It would be much more interesting to start with an brief of mulberry history in China, and point out the important questions that still await answers, then tell the readers that ITS are very good tool to use. Finally, tell clearly to the readers what questions you are going to address.

Second, I found it is very odd that the authors avoid to discuss about morphological species all over the manuscript, and the 33 samples are always named after sampled sites or simply pronunciations of Chinese names. I suppose that the authors did it like this because the molecular data are not consistent with morphological species. Nevertheless, this is one finding of your work, and it should be discussed carefully and logically. It is common that only ITS and cpDNA cannot be able to solve the evolutionary history of cultivated popuations of mulberry, then propose that next generation sequencing data will be a good tool to do that.

Thirdly, according to my experience, the authors are not very familiar with species concept. I recommend the authors read more references about species concept as well as variety and cultivars, e.g. De Queiroz (2007) and references therein, as well as its citations (https://doi.org/10.1080/10635150701701083), and then reconsider their forging of the concept ‘molecular species’.

At last, there are numerous typos and awkwardly presented sentences, I recommend that the authors ask a native English researcher to co-work on this manuscript to improve not only the language, but also the logic and structure of this manuscript that holds considerable potential.

I will be happy to see a revised version of this manuscript, and I am confident that the authors will improve their manuscript to present their findings better.

·

Basic reporting

Line 73-76 The authors introduced former study of classification of Morus with molecular markers, which helped the readers to understand the background. However, the sampling density and the marker used may influence the phylogenetic reconstruction and the classification in the former studies. Thus, it would be helpful for readers to understand the background if the authors mention the number of sampled species in Zhao et al. 2005 and the marker used in Nepal and Ferguson 2012.

Line 79 “grouped into the clade B…”. It would clearer if the authors can add a citation for this “clade B” here, as this classification system has not been introduced in the text above. The authors mainly compared with this classification system (clade B) instead of those listed in line 62-67 in the following discussion. Thus, it is worth to write at least one sentence to summary this system in the introduction part.

I did not find any information about how to access those new sequences in this study. The figures and tables in this article are clear and informative for me, except the tip names of the phylogenetic trees. They are too long to follow except those with scientific names. It would be better if change the code names of tips into M. alba1,2,….

Experimental design

It would be easier for readers to trace the logic of this article if the authors claim their hypotheses or key questions clearly before the brief summary of their results. For now, it is, at least for me, confusing what is the main purpose for this study. I suppose it could be:1) improve the impact of polymorphism of ITS on the phylogenetic reconstruction in Morus; 2) investigating a potential evolutionary pattern (hybridization history) in Morus.

If proposing a new phylogenetic relationship of Morus is one of the key questions of this study, I have one question for the authors: would you please justify the reason for using ITS and two chloroplast markers to reconstruct the phylogenetic relationships? On one hand, the bait set for species level phylogenetic reconstruction in Moraceae has been designed by Gardner et al. (2016 App.PlantSci., see also in Zerega and Gardner 2019 Phytotaxa)., It is worth to try this bait set in Morus, considering the fact that Morus nuclear genome was used as a conference during the design. On the other hand, low-copy nuclear markers have been suggested to be ideal markers for phylogenetic reconstruction in angiosperms (Zhang et al. 2012 NewPhyto). ITS has been proved to show polymorphisms and may mislead the phylogenetic reconstruction in this study, which exactly suggested that ITS is not an ideal marker to do so.

For the “phylogenetic analysis” part in Materials & Methods, I think the authors may have several points to be improved:
1. Outgroups: two Broussonetia species were selected as outgroups. However, Broussonetia and Morus are in different tribes in Moraceae according to the species level phylogenetic reconstruction in the former studies (Clement & Weiblen 2009 Syst. Bot.; Zhang et al. 2019 Ann. Bot.). It would be better to choose phylogenetically closer outgroups to avoid potential long-branch-attraction (Bergsten 2005 Cladistics) and more than one genera would be better.
B. papyrifera alone was used as outgroup in the reconstruction with trnL-trnF and trnT-trnL. However, two species of Broussonetia were selected as outgroups in the reconstruction with ITS. This is not a serious problem, but commonly same species of outgroups will be used in this comparing analyses. Please justify the difference of outgroup selection.
2. Reconstruction approaches: the authors used maximum-likelihood (ML) approach alone to reconstruct the phylogenetic tree, which is the cornerstone of the following discussion of this study. It would be better to apply both Bayesian and ML approach to reconstruct the phylogenetic tree, then compare and confirm the phylogenetic relationship. It is because the advantage and shortcomings of both approaches that we have to reconstruct in this way. In brief, Bayesian approach take into account the uncertainty such as branch length in phylogenetic reconstruction, while applying wrong prior probabilities in Bayesian approach will lead to poor reconstruction (Nei and Kumar, 2000, Molecular evolution and phylogenetics).
3. Substitution model: Kimura 2-parameter and Tamura 3-parameter model were selected to reconstruct the phylogenetic tree. However, the authors did not mention how they make this choice? Which criterion did they use? How many potential models had they compared? Please give more details for model selection.
Line 134 “…two categories (+G,…)” To increase the accuracy of the estimate, the more categories of gamma distribution the better. However, considering the computational burden and the decreasing of accuracy by increasing the number of categories, it is commonly use four or five categories (Nei and Kumar, 2000, Molecular evolution and phylogenetics). Please justify the reason for using two categories.
Another potential problem is reconstruction without considering the presence of invariable sites (+I). Large amount of invariable sites was mentioned in line 246-247 (“The 29 samples with morphological polymorphisms clustered together with a low phylogenetic resoluiton and had similar or identical ITS sequences”). Commonly the presence of invariable sites can be overlooked only when the it is low in dataset. Please justify this point.

Validity of the findings

The authors did not give the detail information such as access number of those sequences they cited from previous studies.

Additional comments

This study provided a phylogenetic analysis of Morus with multiple sampling in four species and single sampling in other twelve species in Morus with two main goals: 1) to attract the attention to the influence of polymorphism in ITS on phylogenetic reconstruction; 2) to investigate the potential evolution processes of Morus. The sampling strategy is a significant improvement over previous studies in Morus. Although several points needed to be improved and confirmed in the phylogenetic reconstruction part, the speculation of the potential evolutionary pattern of Morus is clear and reasonable, which help to extend the understanding of mulberry evolution.

Hereby, I listed some general comments.

Line 75-76 Please keep consistence of the citation. For my understanding, “et al.” is used when the number of authors is more than three and “&” is used when there are only two authors. I suspect “Nepal et al. (line 75)” is the same citation as “Nepal & Ferguson, 2012” in the next line. To keep consistence, it would be better to change the former one into “Nepal and Ferguson”.
Line 80 “among the materials”. I do not understand the meaning of “materials”, do you mean specimens?
Line 82-83 If this “previous study” has been published, please cite it here.
Line 84 It would be better to mention how many species these 33 mulberry samples represented.
Line 151 Loci is different from sites. A locus is a fixed position on a chromosome. A (structure) site is a nucleotide base on a DNA strand. I think “sites” is more accurate in this context.
Line 241 It would be easier to follow if this “clade 3” (including clade 1 and 2) was labelled in Figure 3.
Line 254-256 Phylogenetic reconstruction in genomic scale is also a promising solution for systematic classification of mulberry. It would be a reasonable point in the perspectives in an age of high-throughput sequencing.
Line 284-286 Comparing with which markers trnL-trnF and trnT-trnL are the most variable regions?
Line 303-312 I think there are probably some typos. If I understand well, it must be “pseudogroup 2” in line 303 and 308, and “pseudogroup 1” in line 306. Please capitalize the first letter in “morus” in line 312.

---

## Round 0.2 · Minor Revisions

Please revise the manuscript according to the input of the 3 reviewers.

·

Basic reporting

My conclusion is that this work is just an analysis of cultivated mulberries that has very little practical application to our understanding of the systematics and nomenclature of the genus as a whole. I have no doubt that this is a valuable reference for those working on the breeding of cultivated mulberries and I believe that it would be best to reflect this in the title of the paper where ‘Morus’ should be replaced by ‘cultivated mulberries’. If the presentation could be reorganized to more clearly reflect this, it would then merit publication.

Experimental design

no comment

Validity of the findings

I am still puzzling over how existing names should be applied to the hierarchy of clades presented in this paper. The conclusion states that ‘clade B was separated into two species’ but does not say what these species should be called. This is further confused by Figure 3 which shows 3, not 2, clades within Clade B. Various names are presented at the start of the section on ITS phylogenetic analysis, M. alba, M. wittiorum, M. australis, M. macroura and M. mongolica but these names do not appear within any phylogeny so is their status?
There is also the problem of how to name the two ‘pseudogroups’ in Fig. 3, can these only be referred to as hybrid swarms?

Reviewer 2 ·

Basic reporting

No comment.

Experimental design

No comment.

Validity of the findings

No comment.

Additional comments

This is a manuscript that I reviewed before. I am satisfied with most revision of the authors, which have, together with other referees' comments, improve the clarity and integrity of the story.
Now, two major concern still remains:
(a) The abstract is very difficult to follow by readers who are not familiar with the background, I therefore suggest the authors to further improve to make it more reader-friendly.
(b) Since one of the highlight of this work is the further exploration of Clade B, I suggest the authors to connect Clade B with geographic distribution, key morphology traits or taxonomy, among other factors, if there is any, to make the findings easier to be understood.
Minor concerns:
L39-40: Concerted evolution only act on nrITS sequences, since nrITS is a multiple-copy gene region. It is not proper to say that concerted evolution have played roles in the evlution of mulberry, although it is proper to say that concerted evolution have played roles in the evlution of nrITS in mulberry.
L114: I still do not know what species the authors have sampled for the 33 mulberry accessions. Although the authors said there are seven species in the last part of introduction. Please also give the detail in L114.
L251-L260: I would like to see the morphological species of each accessions, this information is important since accession names does not correlate with any geographical or morphological information!
L271-272: How many accessions of mulberry did the authors find ITS pseudogenes? What is the difference between long/short or polymorphic ITS sequences (14 accessions, L191) and ITS pseudogenes (11 accessions)?
L251: Please use words like "diverged first" among the 33 mulberry accessions. "were the most ancient" is not quite proper in a phylogenetic background, since each accessions kept evolving after divergence.
L434-435: see comments on L39-40 about "concerted evolution".
Table 1: In my opinion, the accessions are not listed in a very reasonable order, I suggest to sort it out following taxa, then the name of accessions alphabetically.

·

Basic reporting

No comment.

Experimental design

No comment.

Validity of the findings

No comment.

Additional comments

This manuscript has been improved substantially after the authors have accepted most of the comments and advices in the first round review. The English is clearer in the new version. However, they still did not claim their scientific questions in the main text, which may confuse the readers. The authors found ITS pseudogenes in muberry and suggested that polyploid, hybridization and concerted evoluiton shaped the evolution of Morus. This study improved our understanding of the evolution of mulberry. I hereby have several minor suggestions.
Line 76, it would be better to mention which genus in Moraceae
Line 83, it would be better to avoid words like “impossible” in a scientific article. It may hard to solve some scientific question for now, or only in the context of this study, but we can hardly say it is “impossible” to solve it, considering the whole history of modern science.
Line 164, Thank you for following my advice in the phylogenetic reconstruction part of Materials and methods. However, it is odd to me that two different criteria were used to select best substitution models in two reconstruction approaches. Please justify this or use the same criterion.
Line 275-284, reads like results for me.
Line 335, please tell the full name of an abbreviation at the first time you mention it.

---

## Round 0.3 · accepted · Accept

Thank you for making the changes suggested by the reviewers. I think your manuscript has been improved considerably.